

# Amelioration of drought effects in wheat and cucumber by the combined application of super absorbent polymer and potential biofertilizer

Yongbin Li[*], Haowen Shi[*], Haowei Zhang and Sanfeng Chen

State Key Laboratory of Agrobiotechnology and College of Biological Sciences, China Agricultural University, Beijing, China

[*] These authors contributed equally to this work.

## ABSTRACT

Biofertilizer is a good substitute for chemical fertilizer in sustainable agriculture, but its effects are often hindered by drought stress. Super absorbent polymer (SAP), showing good capacity of water absorption and retention, can increase soil moisture. However, limited information is available about the efficiency of biofertilizer amended with SAP. This study was conducted to investigate the effects of synergistic application of SAP and biofertilizers (*Paenibacillus beijingensis* BJ-18 and *Bacillus* sp. L-56) on plant growth, including wheat and cucumber. Potted soil was treated with different fertilizer combinations (SAP, BJ-18 biofertilizer, L-56 biofertilizer, BJ-18 + SAP, L-56 + SAP), and pot experiment was carried out to explore its effects on viability of inoculants, seed germination rate, plant physiological and biochemical parameters, and expression pattern of stress-related genes under drought condition. At day 29 after sowing, the highest viability of strain *P. beijingensis* BJ-18 (264 copies ng$^{-1}$ gDNA) was observed in BJ-18 + SAP treatment group of wheat rhizosphere soil, while that of strain *Bacillus* sp. L-56 (331 copies ng$^{-1}$ gDNA) was observed in the L-56 + SAP treatment group of cucumber rhizosphere soil. In addition, both biofertilizers amended with SAP could promote germination rate of seeds (wheat and cucumber), plant growth, soil fertility (urease, sucrose, and dehydrogenase activities). Quantitative real-time PCR analysis showed that biofertilizer + SAP significantly down-regulated the expression levels of genes involved in ROS scavenging (*TaCAT, CsCAT, TaAPX,* and *CsAPX2*), ethylene biosynthesis (*TaACO2, CsACO1,* and *CsACS1*), stress response (*TaDHN3, TaLEA,* and *CsLEA11*), salicylic acid (*TaPR1-1a* and *CsPR1-1a*), and transcription activation (*TaNAC2D* and *CsNAC35*) in plants under drought stress. These results suggest that SAP addition in biofertilizer is a good tactic for enhancing the efficiency of biofertilizer, which is beneficial for plants in response to drought stress. To the best of our knowledge, this is the first report about the effect of synergistic use of biofertilizer and SAP on plant growth under drought stress.

Corresponding author
Sanfeng Chen, chensf@cau.edu.cn

## INTRODUCTION

Traditional agricultural practices have negative influences on continental ecosystem due to huge inputs of chemical fertilizer (*Simonsen et al., 2015*), which has given rise to serious problems such as soil degradation, genetic diversity loss, nitrous oxide emission, and nitrate leaching (*Webb, Harrison & Ellis, 2000*; *Kaur, Brar & Dhillon, 2007*). Biofertilizer containing plant growth-promoting rhizobacteria (PGPR) is an ideal candidate for reducing application of chemical fertilizer in sustainable agriculture (*Rajkumar et al., 2010*; *Choudhury, Kecskes & Kennedy, 2014*). Numerous studies have been reported with respect to growth promotion and biocontrol (*Vrieze et al., 2015*) and colonization of PGPR in different plants. Some PGPR could promote plant growth by enhancing phosphorus (P) solubilization, nitrogen (N) fixation, zinc solubilization, potassium solubilization, and indole-3-acetic acid (IAA) production (*Calvo et al., 2010*; *Khan et al., 2010*; *Islam et al., 2013*; *Gontia-Mishra et al., 2016*). Previous studies have confirmed the roles of *Pseudomonas, Azospirillum, Bacillus, Burkholderia,* and *Paenibacillus* on the growth of pea, strawberry, rice, rape, and maize, respectively (*Chen et al., 2013b*; *Guerrero-Molina et al., 2015*; *Oteino et al., 2015*; *Shakeel et al., 2015*; *Li et al., 2017*). PGPR, such as *Lysinibacillus sphaericus* ZA9, showed great potential in plant growth promotion and biocontrol (*Naureen et al., 2017*). The inoculation with endophytic *P. aeruginosa* PW09 ameliorated both biotic (*Sclerotium rolfsii*) and abiotic (NaCl stress) stresses in cucumber seedlings (*Pandey et al., 2012*). Similarly, the tolerance of chickpea to biotic (*Sclerotinia sclerotiorum*) and abiotic (NaCl stress) stresses was also significantly enhanced after inoculation with *P. putida* S1 or *P. aeruginosa* Cgr (*Ankita et al., 2014*). The application of endophytic *Burkholderia phytofirmans* PsJN mitigated drought stress, thereby improving the growth and development of wheat (*Naveed et al., 2013*). PGPR could also successfully colonize the tissues of plants (*Yang et al., 2013*; *Gontia-Mishra et al., 2016*; *Hao & Chen, 2017*; *Wang et al., 2017*). However, the application of some biofertilizers still remains limitation (*Zhang et al., 2012*; *Grady et al., 2016*). The reason might be that the reproduction of PGPR was affected by adverse climatic conditions (*Zhang et al., 2012*; *Bashan et al., 2013*) such as limited precipitation. Therefore, in arid and semi-arid regions of northern China (*Islam et al., 2010*), drought is a crucial limiting factor, and water shortage hindered the application of biofertilizer in practices.

Drought stress is one of the most destructive environmental factors affecting agricultural production worldwide (*Gontia-Mishra et al., 2016*). Wheat often suffered from periodic drought stress in the growth cycle (*Naveed et al., 2013*; *Naresh, Pramod & Sandeep, 2014*). Super absorbent polymer (SAP), a macromolecular cross-linked and environmentally-friendly polymer, had high capacity of water adsorption and retention (*Johnson & Veltkamp, 1985*), which is conductive to improving seed germination and seedling survival. SAP could be degraded through physical or chemical processes (*Mikkelsen, 1994*). Hence, increasing number of SAP commercial products have been developed to raise water use efficiency, alleviate drought stress, improve soil physical properties and increase crop yield (*Terry, Richard & Nelson, 1986*; *Mikkelsen, 1994*; *Gray, 2011*). Hydrogel supply enhanced soil moisture and seed germination rate (*Rehman, Ahmad & Safdar, 2011*). The application

**Table 1** Bacterial cultures and genomes used in this study.

| Strain No. | Original source | ID | References/Accession No. |
|---|---|---|---|
| BJ-18 | Wheat (rhizosphere isolate), China. | *Paenibacillus beijingensis* | Wang et al. (2013) JN873136 |
| L-56 | Maize (rhizosphere isolate), China | *Bacillus* sp. | MF988362 |

of a hydrophilic polymer (Superab A200) increased dry matter accumulation and water use efficiency of maize (*Dorraji, Golchin & Ahmadi, 2010*). Most studies only focused on the application effect of SAP alone or mixed with a chemical fertilizer, however, to the authors' knowledge, studies on synergistic use of SAP with biofertilizer are rarely reported, particularly under water deficit conditions.

It is of great significance to investigate whether the application of SAP can retain more water and create a suitable environment for reproduction of inoculants.

Soil enzyme activity, as a susceptive indicator of soil health, has potential of comprehensively evaluating soil microbial functional diversity and its functional changes (*Hestrin & Goldblum, 1953*; *Lebrun et al., 2012*). Some soil enzymes participate in N, P, C cycling (*Hestrin & Goldblum, 1953*) and microbial activity. N cycling is mainly governed by urease, protease, and asparaginase; P cycling by alkaline and acidity phosphatase; C cycling by cellulase, sucrase, and β-glucosidase; and microbial activity is regulated by catalase (*Kandeler, Kampichler & Horak, 1996*). *Fusarium* could produce and secrete extracellular enzymes such as phosphatase (*Meyer, Garber & Shaeffer, 1964*). SAP application had little effect on soil microbial metabolism (*Sojka, Entry & Fuhrmann, 2006*), but it still needs more understanding regarding the effect of synergistic use of biofertilizer and SAP on soil enzymes involved in C, N, P cycling, and microbial activity.

In this paper, two important but different crops (wheat: monocotyledon; cucumber: dicotyledon) were selected to comparatively investigate the effects of synergistic use of biofertilizer and SAP on the following aspects: (i) viability of inoculants and soil enzyme activity; (ii) seed germination and plant growth promotion; (iii) plant biochemical indexes and expression of drought stress-related genes under severe drought condition (40% relative soil moisture). Our results are expected to popularize the synergistic use of biofertilizer and SAP, especially in the areas suffered from long-term drought stress.

## MATERIALS & METHODS

### Bacterial strains and potted plant soil

Table 1 details the bacteria used in this study.

The topsoil (0–20 cm depth) was collected from Shangzhuang Experimental Station of China Agricultural University, Beijing, China (40°08′12.15″N, 116°10′44.83″E, 50.21 m above sea level). After air-dried at room temperature, the soil was screened by a 10-mesh sieve to remove plant residues and reduce soil heterogeneity. Then the soil was sterilized at 121 °C for 30 min. The experimental soil were low N-content sandy loam (pH: 7.7, Olsen-P: 7.3 mg kg$^{-1}$, N$_{min}$: 7.8 mg kg$^{-1}$, NH$_4$OAc-K: 115.8 mg kg$^{-1}$, organic matter, 7.2 g kg$^{-1}$).

**Table 2  Description of treatments used for pot experiment.**

| Treatments | Description |
|---|---|
| Control | Rice hull powder without biofertilizer and SAP |
| SAP | Rice hull powder + SAP (100:1, w/w) |
| BJ-18 | Biofertilizer of *P. beijingensis* BJ-18 |
| L-56 | Biofertilizer of *Bacillus* sp. L-56 |
| BJ-18 +SAP | Biofertilizer of *P. beijingensis* BJ-18 + SAP (100:1, w/w) |
| L-56+SAP | Biofertilizer of *Bacillus* sp. L-56 + SAP (100:1, w/w) |

## Preparation of biofertilizer and super absorbent polymer (SAP)

Two PGPR, *P.beijingensis* BJ-18 and *Bacillus* sp. L-56, were inoculated in Erlenmeyer flasks (250 mL) containing 100 mL of Luria Bertani (LB) broth and cultured at 30 °C for 36 h at 180 rpm, respectively. The culture was centrifuged at 6,000 rpm for 5 min and adjusted to $5 \times 10^8$ cells mL$^{-1}$ with sterile normal saline (0.89% w/v NaCl in water). Rice hull was used as carrier, which was ground and screened by a 200-mesh sieve. Then the powder was sterilized at 121 °C for 20 min. The above bacterial suspension was mixed aseptically with sterile rice hull powder (1:1, v/w), and then the mixture was air dried separately in shade as biofertilizer.

The SAP used in this study is polyacrylamide (Dongying Huaye New Material Co., Ltd), which is developed for agriculture and forestry use only. It was sterilized by ultraviolet (UV) light for 1 h in super clean bench (SW-CJ-1F(D)/2F(D), AIRTECH, China), and then mixed with biofertilizer (1:100, w/w) according to the instructions of SAP.

## Plant culture and collection

The experiment was performed to evaluate the role of SAP + biofertilizer under drought condition. Plastic pots (35 cm in diameter; 25 cm in height), which were sterilized by soaking in 0.5 N nitric acid for 24 h, were filled with the above sterilized soil (5 kg). The treatments were listed in Table 2. The mixture (rice hull powder, biofertilizer, and SAP) of each treatment was applied as base manure (9 g/per pot). Before sowing, the pots were watered to 40% relative soil moisture by weighing method. Then, the plants were watered (200 mL each pot) every 5 days till harvest.

## Seed germination assay

For assessment of seed germination, plump seeds of wheat and cucumber were surface sterilized with sodium hypochlorite (10% v/v) for 10 min, followed by rinsing with sterile deionized water. One hundred seeds of each plant were sown in each plastic pot. The experiment design was completely randomized with three replicates for each treatment. The pots were arranged to the greenhouse under optimum condition (15 h day/25 °C–30 °C/day temperature and 9 h night/15 °C–20 °C/night temperature).

The numbers of germinated seeds were recorded daily from 6 days to 14 days after sowing, and the germination rate was calculated according to the reported method (*Sudisha et al., 2010*).

## Viability assessment of inoculants in rhizosphere soil

Two weeks after sowing, the seedlings (wheat and cucumber) were thinned to about 10 cm apart, with only nine uniform and healthy seedlings were left in each pot. At the same time, the mixture (rice hull powder, biofertilizer, and SAP) was applied again according to the description of Table 2 (9 g / per pot, ignoring the weight of SAP). To analyze the viability of inoculants in the rhizosphere soil, the population density was measured by quantitative PCR (qPCR) (*Savazzini et al., 2008*) at day 14, 19, 24, 29, 34, 39, and 44 after sowing. In order to collect the rhizosphere soil, the seedling was uprooted and shaken gently to remove the loosely adhering soil, and the tightly adhering soil regarded as rhizosphere soil. For the qPCR counting, DNA was extracted form rhizosphere soil using an TIANamp Soil DNA Kit (Tiangen Biotech CO., LTD., Beijing, China) according to the manufacturer's protocol. Suitable primers for qPCR were designed based on *nifB* of *P. beijingensis* BJ-18 and *amyE* of *Bacillus* sp. L-56 with the AlleleID 6.01 (PREMIER Biosoft International, Palo Alto CA, USA; Table 3). The 107 bp (*nifB*) and 138 bp (*amyE*) fragments were amplified by conventional PCR and then gel-purified. The PCR products were ligated to PMD 19-T Vector (Takara, Otsu, Japan) and transformed into *Escherichia coli* JM109 competent cells by electrotransformation. The successfully transformed *E. coli* JM109 was cultured in LB liquid medium, afterwards, the plasmids were extracted and purified using a TIANprep Mini Plasmid Kit (TIANGEN BIOTECH(BEIJING) CO., LTD). A standard curve was generated for each run with a dilution range of the recombinant plasmids from $2 \times 10^1$ to $2 \times 10^7$ copies. The above gDNA isolated from different treatments was mixed with the SYBR$^{®}$ Premix Ex Taq$^{TM}$ (Takara, Kyoto, Japan), primer pairs and ddH$_2$O to a total volume of 20 uL for qPCR. Target DNA was quantified based on the above standard curve.

## Sample collection and preparation

The plant and rhizosphere soil samples of wheat and cucumber were harvested at 44 days after sowing. The growth parameters, i.e., root length, shoot length and fresh weight (FW) of each plant, were immediately determined. Four plants per pot were oven dried at 105 °C for 30 min, followed by 65 °C for 72 h. The dry weight (DW) was recorded. Chlorophyll content of seedlings was determined at 8:30–9:30 am on the harvest day using the SPAD-502 chlorophyll meter (Minolta Camera Co. Ltd., Tokyo, Japan). The remaining plant samples were rapidly frozen in liquid N and then stored at −80 °C for further use. Rhizosphere soil samples were screened by a 10-mesh sieve, and then stored in plastic bags for soil enzyme activity determination.

## Determination of free proline content and total soluble sugars (TSS) in plant leaves

Under drought stress, accumulation of osmolytes (free proline content and total soluble sugars) can promote cell growth under adverse osmotic conditions under adverse osmotic conditions. (*Sandhya et al., 2010*). Free proline content in leaves was measured using acid ninhydrin (*Sharma et al., 2010*), and total soluble sugars (TSS) was determined using anthrone reagent (*Shukla, Agarwal & Jha, 2012*).

**Table 3  Primers sequence and accession number in NCBI.**

| Primer | Primer sequence 5′–3′ | NCBI Accession No. | Reference |
| --- | --- | --- | --- |
| nifB | GAAGGTGAGAGTGAGGATGGTTGCTTCAGGCTCATCTCC | MH202771 | This study |
| amyE | CTTCTCGTTCAGGCAGTACTATTGACCGCAGTGATAGC | MH202772 | This study |
| TaCAT | CCATCTGGCTCTCCTACTGGAGAACTTGGACGACGGCCCTGA | E16461 | Moloudi et al. (2013) |
| CsCAT | TTCGGCGTACAAGCATATTCCGTGGCGTGACTGTGATTCG | AY274258 | This study |
| TaAPX | CACCACATCTAAGGGACATCTTCAGAGGGTCACGAGTC | EF555121 | This study |
| CsAPX | ATGATTGTTGACTTGTTATGTGGATGAGGCAGAACGAACC | EU798448 | This study |
| TaNAC2D | ACCTCAGCTACGACGACATCCAGGCGGCGAAGAAGTCATCCGTTCC | GQ231954.1 | Huang & Wang (2016) |
| CsNAC22 | GACCTCGGTCTCGTCTGAAG CTACAATGTTGTTGGACTTCGG | — | Zhang et al. (2017) |
| TaLEA | AGATCGACGGTGACGTGAAGGTCCATGATCTTGCCCAGTAG | — | Li et al. (2018) |
| CsLEA11 | CGAGCAGTTCCAGCTCTACTTCCGGTTAACTTCTCCTT | — | Zhou et al. (2017) |
| TaACO2 | GAGGAACGAGGGCGAGGAG TCAGTTATCAGGCGGTGGC | — | Zhu et al. (2014) |
| CsACO1 | ACCTTCTTCTTACGCCATCGCCACCTACCTTGTCATC | AB006806 | Wei et al. (2015) |
| CsACS1 | CTTCAGGCGTTATTCAGATGGTGCTCGTGTTCTCCATT | U59813.1 | Wei et al. (2015) |
| TaDHN3 | TGGGACGGGCTCAGTGCTATGGGCGGGAGGAGGAAG | — | Zhu et al. (2014) |
| TaPR1-1a | TTCATCATCTGCAGCTACAACC CGGTACATATATACAGCCG-GTCTAA | — | Qi et al. (2012) |
| CsPR1-1a | AACTCTGGCGGACCTTACTCAATATGGCCTTTGGTATAAG | DQ641122 | Dong et al. (2014) |
| TaACTIN | GTCGGTGAAGGGGACTTACATTCATACAGCAGGCAAGCAC | AB181991.1 | Huang & Wang (2016) |
| CsACTIN | AGAGATGGCTGGAATAGAACCTGGTGATGGTGTGAGTC | DQ115883 | Wan et al. (2010) |

## Soil enzyme activity determination

A total of 5 g soil was taken to measure activities of enzymes, which was related to N cycling: urease (EC 3.5.1.5); P cycling: alkaline phosphatase (EC 3.1.3.1) and acid phosphatase (EC 3.1.3.2); C cycling: invertase (EC 3.2.1.26) as well as microbial activity: dehydrogenase (EC 1.1.1). Urease activity was determined with urea as a substrate (*Kaur, Brar & Dhillon, 2007*) and the released N-NH$_4^+$ amount (mg N-NH$_4^+$g soil$^{-1}$ h$^{-1}$) was determined spectrophotometrically at OD$_{578}$. Alkaline phosphatase and acid phosphatase were assessed with *p*-nitrophenyl phosphate as a substrate (*Tabatabai & Bremner, 1969*) and the released *p*-nitrophenol (pNP) amount (mg pNP g soil$^{-1}$ h$^{-1}$) was measured spectrophotometrically at OD$_{660}$. Sucrase activity was measured with sucrose as a substrate (*Chen et al., 2013a*) and the released glucose amount (mg glucose g soil$^{-1}$ h$^{-1}$) was determined spectrophotometrically at OD$_{508}$. Dehydrogenase activity was determined with 2,3,5-triphenyltetrazolium chloride (TTC) as a substrate (*Schinner et al., 1996*) and the reduced triphenylformazan (TPF) amount (mg TPF g soil$^{-1}$ h$^{-1}$) was determined spectrophotometrically at OD$_{485}$. All assays were carried out in triplicate.

## Quantitative real-time (qRT) PCR analysis of drought stress-related genes

Total RNA was extracted from leaf samples of wheat and cucumber using Trizol reagent according to the manufacturer's protocol. RNA concentration was determined at 260 nm using a spectrophotometer (Nanodrop 1000; Thermo Scientific, Waltham, MA, USA).

Strand cDNA was synthesized using PrimeScript$^{TM}$ RT reagent kit (Takara, Kyoto, Japan) following the instructions of the manufacturer. Then, cDNA was used as template to determine gene expression levels. qRT-PCR was performed by using the SYBR$^{®}$ Premix Ex Taq$^{TM}$ (Takara, Kyoto, Japan). The specific primers of *ACTIN* were used as reference. The specific primers used for qRT-PCR were designed by using the AlleleID 6.01 (PREMIER Biosoft International, Palo Alto, CA, USA) or from previous literature (Table 3). The relative expression levels were calculated according to the standard comparative $C(t)$ method (*Livak & Schmittgen, 2001*). Each treatment had three biological replicates, with three technical replicates for each biological replicate.

## Statistical analysis

All data was statistically analyzed using SPSS software version 20 (SPSS Inc., Chicago, IL, USA). The significant differences among treatments were analyzed using one-way analysis of variance (ANOVA) followed by least significant difference (LSD) at 0.05 level of probability. Graphs were prepared with SigmaPlot software version 12.5 (Systat Software, Inc, San Jose, CA, USA).

# RESULTS

## Effects of biofertilizer and SAP on seed germination

The number of germination seeds was dynamically observed from day 5 to day 14 after sowing (Table 4). The synergistic use of biofertilizer and SAP had significant effects on germination rate and germination time of wheat and cucumber seeds. The germination time of wheat seeds treated with biofertilizer + SAP shortened by 4 days compared with that of the control, while 2 days compared with that of SAP treatment group. The germination rate of wheat seeds was 22.70% in the control; while 70.0% in BJ-18 + SAP treatment group and 62.0% in L-56 + SAP treatment group. The results indicated that adding SAP to the biofertilizer of *P. beijingensis* BJ-18 and *Bacillus* sp. L-56 increased the seed germination rate by 47.3% and 39.3%, respectively, compared with the control.

Similarly, biofertilizer + SAP treatment group shorten the germination time of cucumber seed by 2–4 days compared with other treatment groups. Among all the treatments of cucumber, L-56 + SAP treatment group had the highest germination rate, with a 35.7% increase compared to the L-56 treatment group.

## Effect of biofertilizer and SAP on viability of inoculant

The qPCR method was used to determine the inoculant population density in the rhizosphere soil, and the viability of inoculants was significantly enhanced when biofertilizer was applied with SAP as compared to other treatments (Table 5). The inoculant populations increased at first and then declined. In wheat rhizosphere, the strain of BJ-18 + SAP treatment group showed the highest bacterial population (264 copies ng$^{-1}$ gDNA) at day 29 after sowing. After reaching the highest population, both strains of *P. beijingensis* BJ-18 and *Bacillus* sp. L-56 showed a continuous declining trend. In cucumber rhizosphere, the L-56 + SAP treatment group showed the highest bacterial population (331.7 copies ng$^{-1}$ gDNA) at day 29; and retained a higher level (75.0 copies ng$^{-1}$ gDNA) till day 44 than

**Table 4** **Effects of super absorbent polymer supply on germination of wheat and cucumber seeds.**

| Plant | Treatments | Seed germination (%) at different day | | | | | | | | | |
|---|---|---|---|---|---|---|---|---|---|---|---|
| | | 5 D | 6 D | 7 D | 8 D | 9 D | 10 D | 11 D | 12 D | 13 D | 14 D |
| Wheat | Control | $0^a$ | $0^a$ | $0^b$ | $0^c$ | $0^d$ | $0^c$ | $6.3 \pm 1.2^d$ | $9.7 \pm 0.3^d$ | $18.0 \pm 1.0^e$ | $22.7 \pm 1.3^e$ |
| | SAP | $0^a$ | $0^a$ | $0^b$ | $0^c$ | $5.7 \pm 1.2^c$ | $12.3 \pm 0.9^b$ | $24.3 \pm 1.5^b$ | $37.7 \pm 1.8^b$ | $48.0 \pm 2.3^c$ | $52.3 \pm 3.0^c$ |
| | BJ-18 | $0^a$ | $0^a$ | $0^b$ | $0^c$ | $0^d$ | $1.0 \pm 1.0^c$ | $7.7 \pm 0.7^c$ | $17.7 \pm 1.2^c$ | $29.0 \pm 1.2^d$ | $33.0 \pm 0.6^d$ |
| | L-56 | $0^a$ | $0^a$ | $0^b$ | $0^c$ | $0^d$ | $1.0 \pm 1.0^c$ | $7.7 \pm 0.7^{cd}$ | $13.0 \pm 1.7^{cd}$ | $22.7 \pm 1.4^{de}$ | $29.7 \pm 1.3^d$ |
| | BJ-18 +SAP | $0^a$ | $0^a$ | $8.7 \pm 1.3^a$ | $14.7 \pm 2.1^a$ | $20.3 \pm 2.0^b$ | $46.0 \pm 2.5^a$ | $50.7 \pm 1.8^a$ | $56.3 \pm 2.3^a$ | $67.7 \pm 1.8^a$ | $70.0 \pm 1.0^a$ |
| | L-56+SAP | $0^a$ | $0^a$ | $6.7 \pm 2.0^a$ | $17.3 \pm 2.5^b$ | $24.7 \pm 1.2^a$ | $44.3 \pm 2.4^a$ | $52.3 \pm 2.2^a$ | $57.7 \pm 7.8^a$ | $60.7 \pm 1.4^b$ | $62.0 \pm 2.1^b$ |
| Cucumber | Control | $0^a$ | $0^a$ | $0^a$ | $0^b$ | $0^c$ | $0^c$ | $0^d$ | $4.3 \pm 0.8^c$ | $6.3 \pm 0.7^d$ | $14.3 \pm 0.9^c$ |
| | SAP | $0^a$ | $0^a$ | $0^a$ | $0^b$ | $0^c$ | $13.3 \pm 2.0^b$ | $17.3 \pm 2.2^b$ | $21.7 \pm 3.9^b$ | $31.3 \pm 4.1^b$ | $39.3 \pm 3.3^b$ |
| | BJ-18 | $0^a$ | $0^a$ | $0^a$ | $0^b$ | $0^c$ | $1.0 \pm 1.0^c$ | $5.3 \pm 1.2^c$ | $8.7 \pm 1.2^c$ | $12.0 \pm 1.5^{cd}$ | $19.3 \pm 1.2^c$ |
| | L-56 | $0^a$ | $0^a$ | $0^a$ | $0^b$ | $0^c$ | $0^c$ | $4.3 \pm 0.9^{cd}$ | $7.3 \pm 2.0^c$ | $13.7 \pm 1.2^c$ | $20.3 \pm 1.3^c$ |
| | BJ-18 +SAP | $0^a$ | $0^a$ | $0^a$ | $8.3 \pm 2.1^a$ | $10.0 \pm 1.0^b$ | $27.7 \pm 3.8^a$ | $35.0 \pm 1.5^a$ | $37.7 \pm 2.2^a$ | $44.7 \pm 2.0^a$ | $51.3 \pm 1.8^a$ |
| | L-56+SAP | $0^a$ | $0^a$ | $0^a$ | $8.3 \pm 3.1^a$ | $13.0 \pm 1.0^a$ | $24.3 \pm 2.3^a$ | $32.7 \pm 2.0^a$ | $41.7 \pm 2.6^a$ | $49.7 \pm 2.8^a$ | $56.0 \pm 3.2^a$ |

**Notes.**

Values are given as mean ± SE of three independent biological replicates, and bearing different letters (a, b, c) at one specific time (e.g., 5 or 6 days) are significantly different from each other according to the least significant difference (LSD) test ($p < 0.05$).

D, day; SAP, Super Absorbent Polymer; BJ-18, *P. beijingensis* strain BJ-18; L-56, *Bacillus* sp. strain L-56.

**Table 5  Survival of different bacterial strains inoculated to wheat and cucumber and population dynamics in the rhizosphere after thinning.**

| Plant | Treatments | Inoculant concentration in the rhizospher (copies ng$^{-1}$ gDNA) | | | | | | |
|---|---|---|---|---|---|---|---|---|
| | | 14 D | 19 D | 24 D | 29 D | 34 D | 39 D | 44 D |
| Wheat | BJ-18 | 123.9 ± 8.1[b] | 142.0 ± 4.0[d] | 140.0 ± 1.0[d] | 124.2 ± 3.9[d] | 96.1 ± 3.2[c] | 37.1 ± 1.6[c] | 19.3 ± 0.2[b] |
| | L-56 | 198.9 ± 20.0[a] | 214.5 ± 0.9[b] | 200.2 ± 1.4[b] | 180.7 ± 5.8[b] | 130.3 ± 1.4[b] | 48.6 ± 3.7[c] | 17.2 ± 0.2[b] |
| | BJ-18 +SAP | 121.2 ± 1.6[b] | 157.7 ± 3.7[c] | 173.1 ± 3.2[c] | 264.4 ± 14.7[c] | 184.2 ± 4.6[a] | 109.8 ± 7.9[a] | 68.5 ± 4.3[a] |
| | L-56+SAP | 210.2 ± 1.9[a] | 236.6 ± 2.4[a] | 254.4 ± 1.0[a] | 237.7 ± 2.0[a] | 182.9 ± 2.1[a] | 140.3 ± 1.4[a] | 61.9 ± 6.7[a] |
| Cucumber | BJ-18 | 110.7 ± 3.3[c] | 129.1 ± 1.5[d] | 127.2 ± 4.9[c] | 116.3 ± 12.1[d] | 83.6 ± 5.4[c] | 53.8 ± 3.9[c] | 9.7 ± 0.6[d] |
| | L-56 | 184.0 ± 12.9[b] | 195.2 ± 1.8[b] | 156.8 ± 5.2[b] | 144.5 ± 5.7[c] | 99.8 ± 10.4[c] | 45.1 ± 2.9[c] | 18.4 ± 0.3[c] |
| | BJ-18 +SAP | 116.4 ± 2.6[c] | 154.8 ± 8.4[c] | 161.7 ± 3.2[b] | 188.0 ± 6.5[b] | 158.7 ± 10.6[b] | 94.5 ± 2.6[b] | 54.0 ± 2.6[b] |
| | L-56+SAP | 213.0 ± 3.9[a] | 240.9 ± 3.2[a] | 317.2 ± 2.1[a] | 331.7 ± 2.5[a] | 240.9 ± 11.8[a] | 113.5 ± 6.0[a] | 75.0 ± 2.8a |

**Notes.**
Values are given as mean ± SE of three independent biological replicates, and bearing different letters (a, b, c) at one specific time (e.g., 14 or 19 days) are significantly different from each other according to the least significant difference (LSD) test ($p < 0.05$).

D, day; SAP, Super Absorbent Polymer; BJ-18, *P. beijingensis* strain BJ-18; L-56, *Bacillus* sp. strain L-56.

others. No such inoculants (*P. beijingensis* BJ-18 and *Bacillus* sp. L-56) populations were detected in the rhizosphere soil of un-inoculated seedlings (wheat and cucumber).

## Effect of biofertilizer and SAP on plant growth parameters

The treatments of adding SAP to biofertilizer significantly increased the growth parameters of both wheat and cucumber seedlings, including plant length, FW and DW (Fig. 1). The BJ-18 + SAP treatment group showed maximum increase over control in shoot length (28.2%), root length (42.3%), shoot FW (86.9%), root FW (83.4%), shoot DW (104.9%) and root DW (79.7%) of wheat seedlings. However, no significant effect was found in BJ-18 treatment group and L-56 treatment group in comparison to control.

In case of cucumber plants, the treatment group of L-56 + SAP showed maximum increase over control in shoot length (47.8%), root length (50.8%), shoot FW (71.9%), root FW (175.7%), shoot DW (68.9%) and root DW (76.9%). All the biometric growth parameters showed an order of L-56 + SAP >BJ-18 + SAP >L-56 >BJ-18 >Control. Similarly, cucumber treated with biofertilizer alone had no significant difference in all the biometric growth parameters as compared to control.

## Effect of biofertilizer and SAP on biochemical parameters of seedlings

For both wheat and cucumber, the biochemical parameters (proline content, TSS content, and chlorophyll content) of leaves were quantified (Fig. 2). The results indicated that L-56 + SAP treatment group showed the lowest proline content, followed by BJ-18 + SAP group. The seedlings also showed a significant decline in total soluble sugar content in contrast to corresponding control. The chlorophyll content was expressed as SPAD value. A significant increase in chlorophyll content of both wheat and cucumber seedlings was observed in the BJ-18 + SAP treatment group compared to corresponding control (SPAD value of 59.0 and 46.3, respectively).

## Effect of biofertilizer and SAP on soil extracellular enzyme activities

The activities of extracellular enzymes (urease, sucrose, acid phosphate, alkaline phosphate, and catalase) in rhizosphere soil were determined (Fig. 3). In the wheat and cucumber

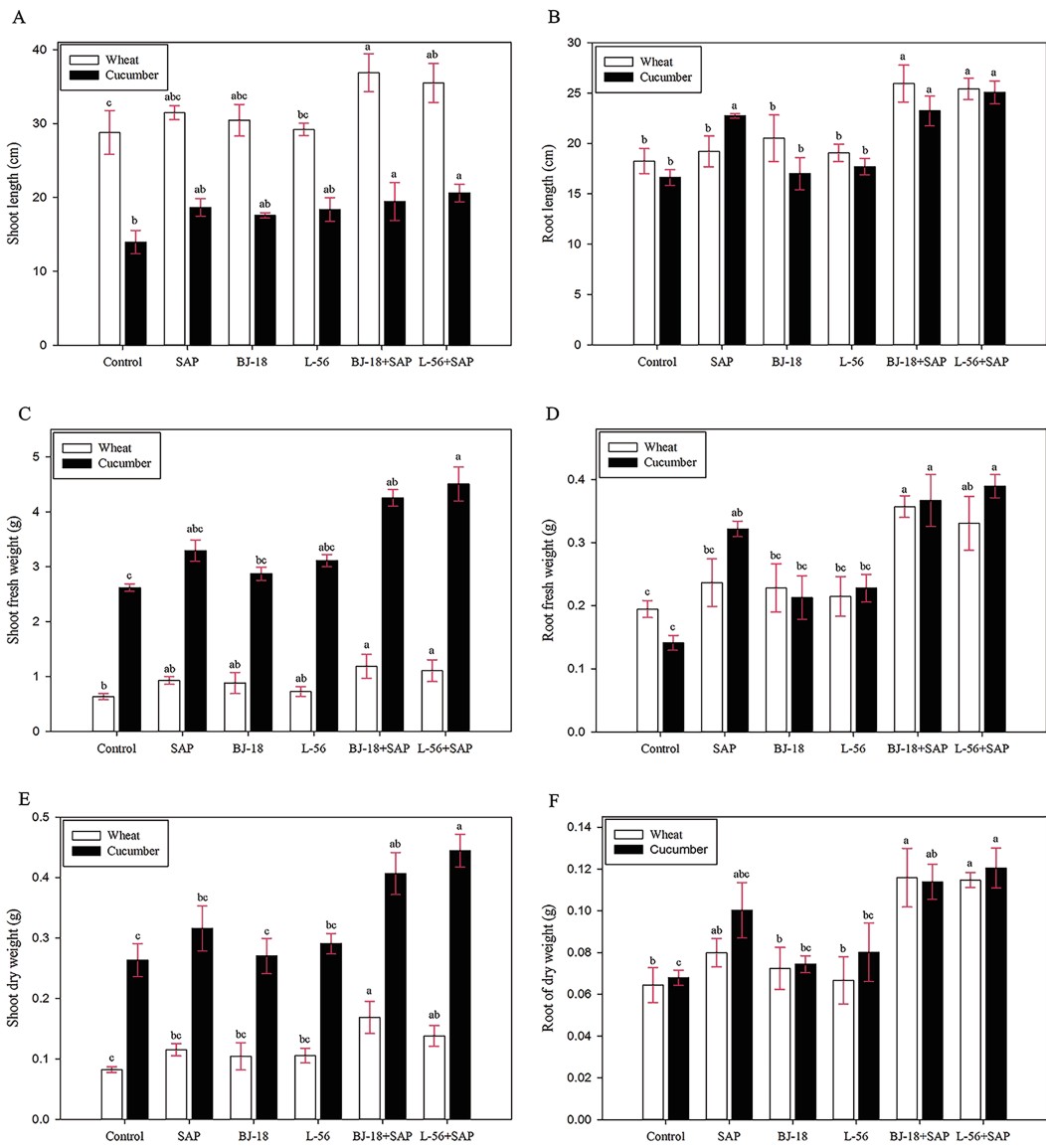

**Figure 1** **Effects of super absorbent polymer supply on shoot length (A), root length (B), shoot fresh weight (C), root fresh weight (D), shoot dry weight (E) and root dry weight (F) of wheat and cucumber seedlings.** Values are given as mean of three independent biological replicates, and bearing different letters (a, b, c) are significantly different from each other according to the least significant difference (LSD) test ($p < 0.05$). The bars represent the standard error. SAP: Super Absorbent Polymer, BJ-18: *P. beijingensis* strain BJ-18, L-56: *Bacillus* sp. strain L-56.

rhizosphere soil, urease activity in the BJ-18 + SAP group was greater than others (Fig. 3A). The BJ-18 + SAP treatment had significant positive effects on sucrose activity of the wheat rhizosphere soil as compared to control (Fig. 3B). However, the L-56 + SAP group showed the lowest sucrose activity in the cucumber rhizosphere soil (Fig. 3B). It was observed that the activity of alkaline phosphatase in different treatments was higher than that of acid phosphatase, and all treatment groups showed no significant difference in both acid

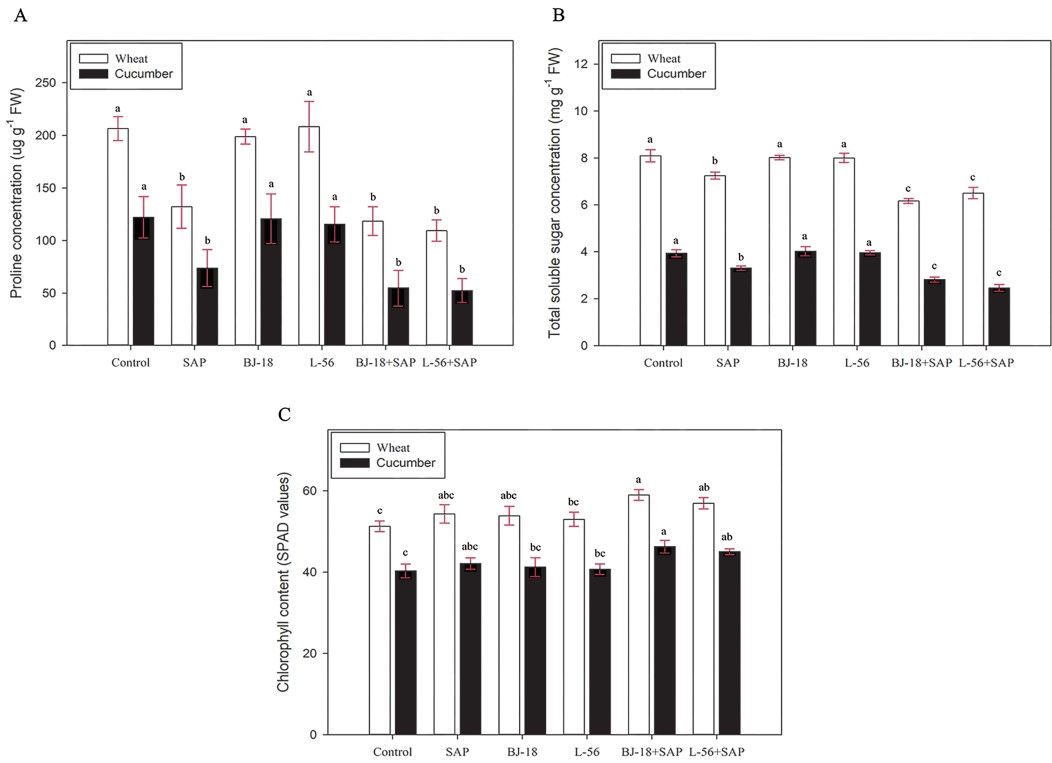

**Figure 2** **Effects of super absorbent polymer supply on proline concentration (A), total soluble sugar concentration (B), chlorophyll content (C) of wheat and cucumber seedlings.** Values are given as mean of three independent biological replicates, and bearing different letters (a, b, c) are significantly different from each other according to the least significant difference (LSD) test ($p < 0.05$). The bars represent the standard error. SAP: Super Absorbent Polymer, BJ-18: P. beijingensis strain BJ-18, L-56: *Bacillus* sp. strain L-56.

and alkaline phosphatase activities in comparison to corresponding control (Figs. 3C and 3D). In the Fig. 3E, the dehydrogenase activity was the lowest in BJ-18 + SAP group of wheat rhizosphere soil, while it was the lowest in the L-56 + SAP group of cucumber rhizosphere soil.

## Effect of PGPR and SAP on expression of stress-responsive genes

The qRT-PCR results showed the drought stress-related genes had different expression patterns under drought stress in biofertilizer + SAP treatment group and SAP treatment group (Fig. 4). The transcript levels of *TaCAT, TaAPX, TaACO, TaDHN, TaLEA, TaPR1,* and *TaNAC* were down-regulated in wheat treated with BJ-18+SAP and L-56 + SAP as compared to the control.

Variability in expression was also observed in the cucumber seedlings. The transcript levels of *CsCAT, CsAPX, CsACO, CsACS, CsLEA, CsPR1, and CsNAC* were down-regulated in cucumber treated with BJ-18+SAP and L-56 + SAP as compared to the control

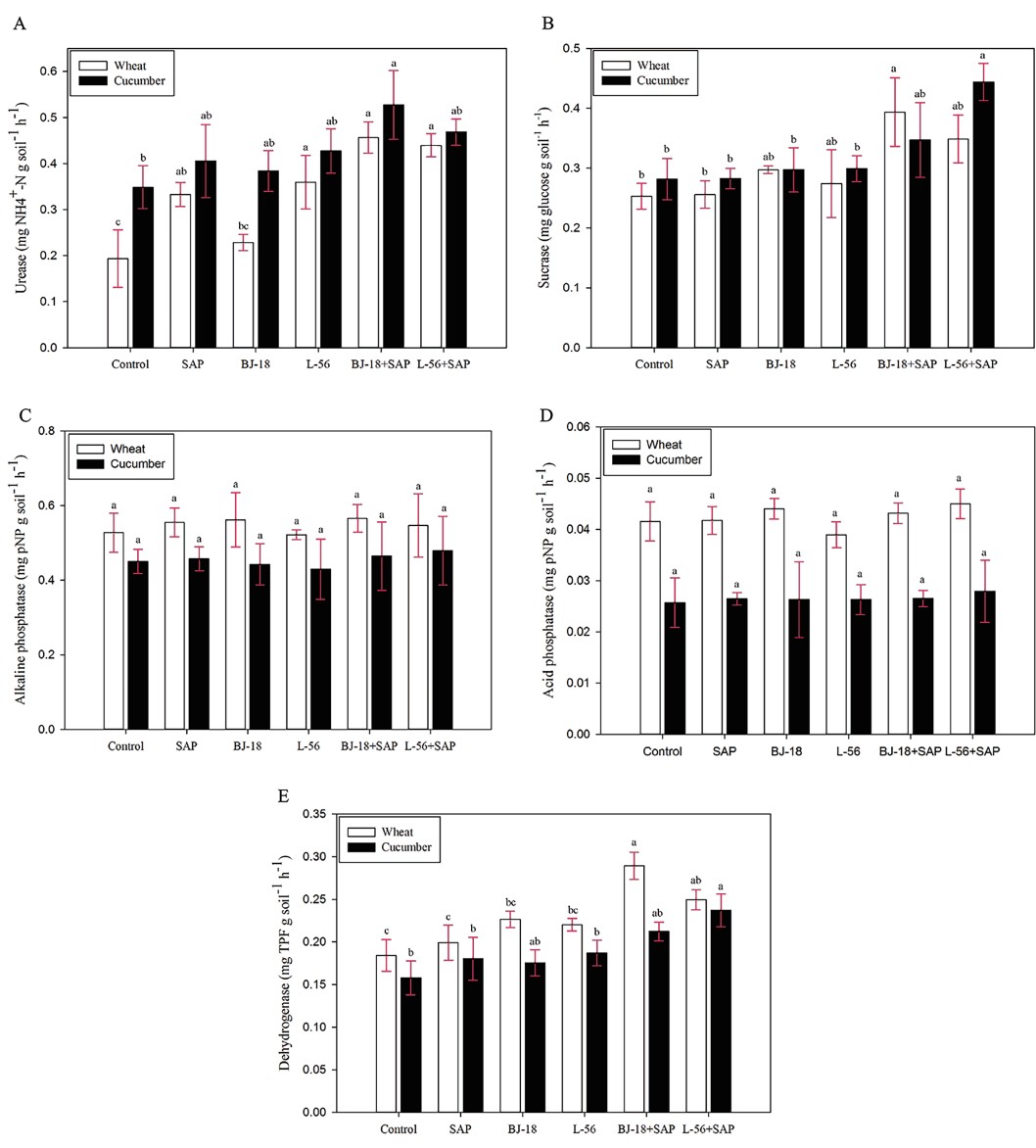

**Figure 3** **Effects of super absorbent polymer supply on enzyme activities of urease (A), sucrose (B), alkaline phosphatase (C), acid phosphatase (D) and dehydrogenase (E) of rhizosphere soil.** The values are given as mean of three independent biological replicates, and bearing different letters (a, b, c) are significantly different from each other according to the LSD test ($p < 0.05$). The bars represent the standard error. SAP: Super Absorbent Polymer, BJ-18: *P. beijingensis* strain BJ-18, L-56: *Bacillus* sp. strain L-56, pNP: *p*-Nitrophenol, TPF: Triphenylformazan.

# DISSCUSSION

Water deficiency suppresses PGPR reproduction of biofertilizer in the arid and semiarid areas of North China, and the application of biofertilizer was limited because no positive effect on crop growth and yield was observed. This study was aimed at minimizing the negative effect of water deficiency on biofertilizer by adding SAP to biofertilizer. The

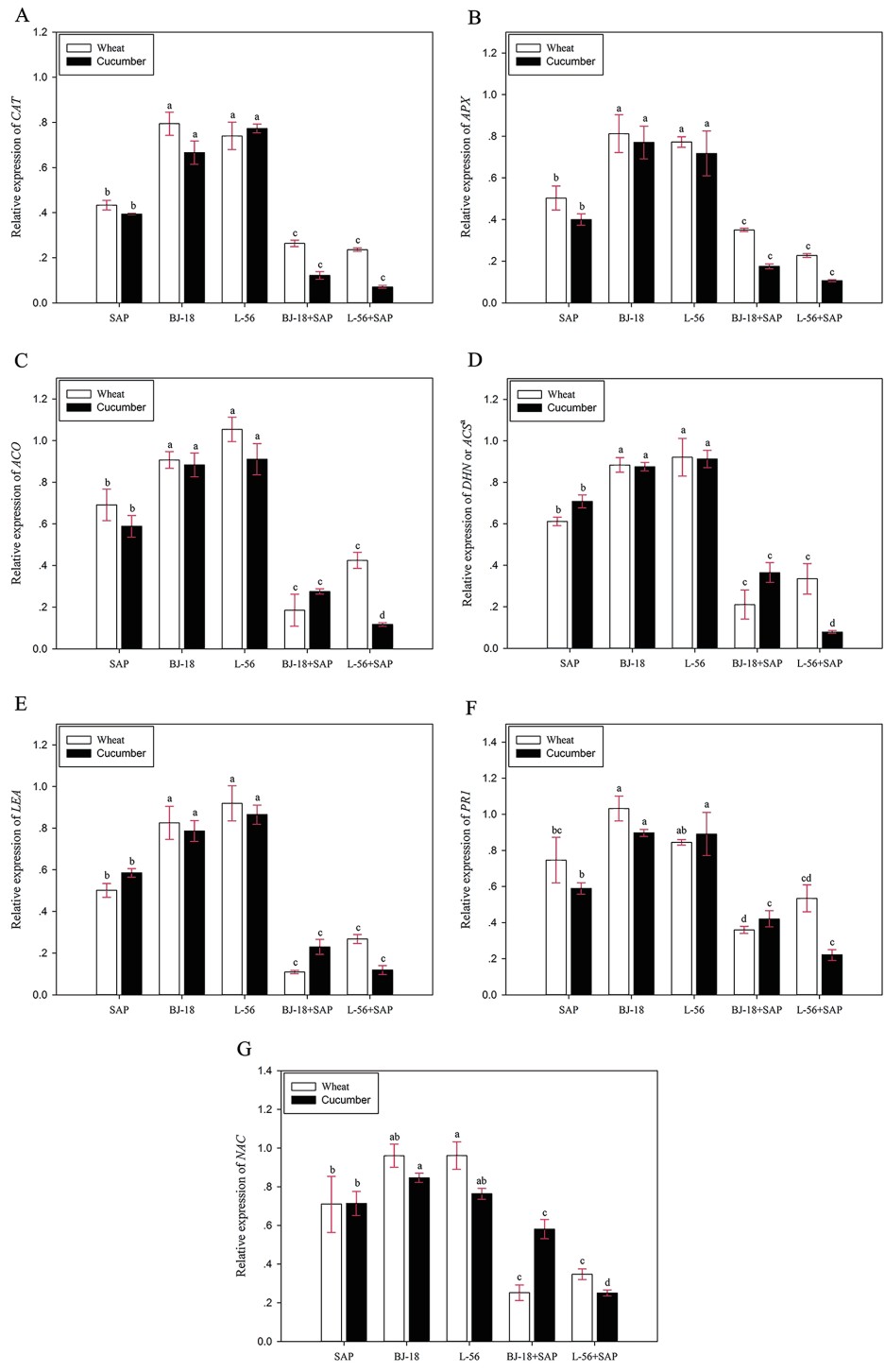

**Figure 4  Effects of super absorbent polymer supply on relative gene expression level of stress-genes in wheat and cucumber seedlings.** The values are given as mean of three independent biological replicates, and bearing different letters (a, b, c) are significantly different from each other according to the least significant difference (LSD) test ($p < 0.05$). The bars represent the standard error. SAP: Super Absorbent Polymer, BJ-18: *P. beijingensis* strain BJ-18, L-56: *Bacillus* sp. strain L-56.. a: *DHN* for wheat, *ACS* for cucumber.

results revealed that synergistic use of biofertilizer and SAP decreased adverse effect of water deficiency on PGPR, and improved the efficiency of biofertilizer.

Germination is one of the most vital stages in a crop life cycle, which is significantly impacted by drought stress (*Sleimi et al., 2013*). Both the SAP treatment group and the PGPR + SAP treatment group significantly enhanced the germination rate of wheat and cucumber seeds. The use of SAP could improve the water-holding capacity of soil (*Johnson, 1984*), which is beneficial to seed germination. The application of SAP improved *Caragana korshinskii* seed germination rate (*Su et al., 2017*). Inoculation with PGPR (*P. fluorescens*, *Enterobacter hormaechei*, and *P. migulae*) could promote foxtail millet seed germination and seedling growth under drought condition (*Niu et al., 2018*). PGPR of *Burkholderia* sp. L2 and *Bacillus* sp. A30 could increase tomato seed germination (*Tripti et al., 2017*). In addition, the inoculation of *Pseudomonas*, *Azospirillum,* and *Azotobacter* significantly improve maize seed germination (*Gholami, Shahsavani & Nezarat, 2009*). In this study, the SAP application created an optimal environment for PGPR reproduction and seed germination by increasing soil moisture. Therefore, both BJ-18 + SAP and L-56 + SAP groups presented the maximum germination rate of wheat and cucumber.

The survival and persistence of PGPR introduced into rhizospher soil is a most vital factor for successful application of biofertilizer (*Naqqash et al., 2016*). Here, the survival and persistence of inoculants (*P. beijingensis* BJ-18 and *Bacillus* sp. L-56) in the rhizosphere soil was dynamically observed during the period of day 14–44 using qPCR. All groups except control exhibited consistent trend in populations of inoculants, which were increased firstly and declined afterwards. Similar trend has been observed in the strain of *Phomopsis liquidambari* (*Wang et al., 2014*). In general, the combination use of biofertilizer and SAP significantly increased the bacterial populations compared with corresponding biofertilizer use alone, in both wheat and cucumber rhizosphere soil. Even at day 44, the largest bacterial population was observed in BJ-18 + SAP group of wheat rhizosphere (68.5 copies ng$^{-1}$ gDNA), and in L-56 + SAP group of cucumber rhizosphere (75.0 copies ng$^{-1}$ gDNA). Numerous studies have reported the populations of introduced microorganisms were significantly reduced in the natural soil (*Pujol et al., 2006*; *Kumar, Trivedi & Pandey, 2007*; *Longa et al., 2009*) due to the hostility of adverse biotic factor (competition with native organisms for finite nurture) and abiotic factors (temperature, humidity, pH values) within the soil (*Avrahami & Bohannan, 2007*). Our results indicated that adding SAP to biofertilizer could minimize the adverse effect of water deficiency, thus improving the inoculant viability and prolonging persistence. It has been reported that the population density of *Pseudome fluorescents* in biochar and/or compost treated soil significantly higher than that in un-treated soil, which is achieved by increasing soil moisture (*Nadeem et al., 2017*).

In wheat and cucumber, both BJ-18 group and L-56 group did not significantly improve plant growth parameters (length, FW, DW) as compared to the control, which was not consistent with the results of previous studies, where *Klebsiella* sp. MBE02, *Paenibacillus* sp. NSY50, *Bacillus,* and *Pseudomonas* sp. MS16 enhanced the growth of peanut, cucumber, rice, wheat and respectively (*Sharma, Kulkarni & Jha, 2016*; *Du et al., 2016*; *Feng et al., 2017*; *Muhammad et al., 2018*). However, with the combination use of SAP (BJ-18 + SAP
group and L-56 + SAP group), the above parameters (length, FW, DW) were significantly elevated in both wheat and cucumber seedlings as compared to the corresponding control. Particularly, BJ-18 + SAP treatment significantly promoted the root length, root FW, shoot DW and root DW compared with the SAP treatment in wheat, and also increased shoot FW, shoot FW and shoot DW in cucumber; L-56 + SAP treatment significantly promoted the shoot FW, root FW and shoot DW in wheat compared with the SAP treatment, and also increased shoot FW and shoot DW in cucumber. Considering the water-holding capacity of SAP, it was hypothesized that SAP is conductive to creating a more suitable environment for the reproduction of introduced inoculants in soil under drought stress. Such hypothesis was proven by the fact that survival of *P. beijingensis* BJ-18 and *Bacillus* sp. L-56 was greater in SAP treated soil as compared to the un-amended soil.

After application of biofertilizer amended with SAP in wheat and cucumber, the activities of urease, sucrose and dehydrogenase were significantly increased as compared the control. Urease increased utilization of N fertilizer by catalysing urea hydrolysis into ammonia (*Bowles et al., 2014*). No matter for wheat or cucumber, the BJ-18 + SAP group showed the strongest urease activity in all treatments, which might be related to the N-fixing ability of *P. beijingensis* BJ-18 (*Wang et al., 2013*). The activity of dehydrogenase, a vital indicator of microorganism activity, was increased after the application of inoculation to soil (*Siddikee et al., 2016*). Here, the highest activity of dehydrogenase and sucrose was detected in the BJ-18 + SAP group of wheat rhizosphere soil and L-56 + SAP group of cucumber rhizosphere soil, which was consistent with the survival of *P. beijingensis* BJ-18 and *Bacillus* sp. L-56 in the wheat or cucumber rhizosphere soil. These findings were also consistent with previous report (*Xun et al., 2015*), which demonstrates that inoculation with PGPR could significantly enhance the activities of soil enzymes (urease, dehydrogenase, and sucrose). However, acid phosphatase (alkaline phosphatase) activity showed no significant difference among different treatments, indicating that these two inoculants have little effects on this enzyme. In addition, the activity of alkaline phosphatase in different treatments was higher than that of acid phosphatase, which might be related to the influence of alkaline soil. Similar results have been reported in rice inoculated with *Phomopsis liquidambari* under LN level (*Siddikee et al., 2016*).

Under drought stress, the accumulation of proline and total soluble sugar can protect plant cells from osmotic damage (*Delauney & Verma, 1993*; *Gontia-Mishra et al., 2016*; *Tiwari et al., 2016*). Accordingly, a reduction in proline and total soluble sugar content was observed in both wheat and cucumber seedlings in the SAP treatment group compared with control, BJ-18 and L-56 treatments. However, wheat and cucumber showed a further decrease in proline and total soluble sugar in the BJ-18 + SAP and L-56 + SAP treatments as compared with the SAP treatment group. It was obvious that SAP amendment could help the biofertilizer to perform better by providing a suitable habitat for inoculants in soil, and therefore, less proline and total soluble sugar were accumulated by synergistic use of biofertilizer and SAP. Under drought stress, less proline accumulation of *Caragana korshinskii* grown in SAP amended soil was also reported by (*Su et al., 2017*). Chlorophyll content is also significant biochemical indicator of stress tolerance in plants (*Percival, Fraser & Oxenham, 2003*). In this study, the BJ-18 + SAP and L-56 + SAP treatments

significantly improved the chlorophyll content in the wheat and cucumber seedlings. The application of biochar could also increase the chlorophyll content of *Chenopodium quinoa* by improving soil moisture content (*Gorim & Asch, 2017*). PGPR-inoculated plants also showed a significant increase in chlorophyll content (*Kakar et al., 2016*; *Abdelkrim et al., 2018*; *El-Esawi et al., 2018*).

To better understand how the plants (wheat and cucumber) respond to the different treatments at the transcriptional level, some representative drought-responsive genes were selected. Both *CAT* and *APX* genes were involved in reactive oxygen species (ROS) scavenging, and ROS overproduction resulted in a negative oxidative stress on plant growth. (*Lata et al., 2011*; *Tiwari et al., 2016*). qRT-PCR analysis showed that under drought stress, the expression levels of *APX* and *CAT* in seedlings of the biofertilizer + SAP groups were lower than those of other treatment groups, suggesting that biofertilizer + SAP treatments could relieve drought stress and ensure normal growth of seedlings. It was also reported chickpea inoculated with *Pseudomonas* showed significantly lower expression of *CAT* and *APX* during the stress period (*Tiwari et al., 2016*; *Saif & Khan, 2018*). Ethylene has been reported to regulate some different aspects of plant growth and development, particularly abiotic stressed such as drought stress (*Yang, Kloepper & Ryu, 2009*; *Habben et al., 2014*). Both *ACO* and *ACS* are involved in ethylene metabolism. In this study, the biofertilizer + SAP treatments significantly repressed the expression levels of *CAT* and *APX*, indicating less ethylene accumulation in plants treated with biofertilizer + SAP. Dehydrin and late embryogenesis abundant protein encoded by *DHN* and *LEA* genes play great roles in adaptive responses of plants to drought stress (*Gao et al., 2008*). The biofertilizer + SAP groups showed relatively low expression level of *DHN* and *LEA*. There were plenty of evidence for the involvement of salicylic acid in plant drought stress (*Fujita et al., 2006*). *PR1* participate in salicylic acid metabolism. In this study, the biofertilizer + SAP treatments significantly inhibited the expression of *PR1*. NAC transcription factors have been reported to participate in biotic and abiotic stress tolerance in plants (*Delessert et al., 2005*; *Nakashima, Ito & Yamaguchi-Shinozaki, 2009*; *Wu et al., 2009*). *TaNAC2D* was induced by abiotic stress (*Huang & Wang, 2016*) and *CsNAC22* responded to drought stress (*Zhang et al., 2017*). In this study, the biofertilizer + SAP groups inhibited the expression of *NAC*. Taken together, our results suggest that the biofertilizer + SAP treatments significantly ameliorate drought stress.

## CONCLUSIONS

This study showed that the addition of SAP significantly enhanced survival rate of inoculants (*P. beijingensis* BJ-18 and *Bacillus* sp. L-56), and then promoted seed germination of wheat and cucumber, plant growth, soil fertility (urease, sucrose and dehydrogenase). qRT-PCR analysis also showed that the transcript levels of some stress-related genes were down-regulated in wheat and cucumber treated with biofertilizer + SAP, respectively, which imply that the biofertilizer + SAP treatments contribute to drought tolerance of wheat and cucumber. Our results indicate that SAP addition in biofertilizer is a good strategy for improving the efficiency of biofertilizer, especially in the areas suffering from long-term

drought stress. This study was carried out in the greenhouse; therefore, further experiments are still needed to confirm the effect of biofertilizer amended with SAP in the field.

## ACKNOWLEDGEMENTS

We would like to thank Caixia Wang for helping to improve manuscript.

### Funding

This work was supported by the National Key Research and Development Program of China (No. 2017YFD0200807). The funders had no role in study design, data collection and analysis, decision to publish, or preparation of the manuscript.

### Grant Disclosures

The following grant information was disclosed by the authors:
National Key Research and Development Program of China:  No. 2017YFD0200807.

### Competing Interests

The authors declare there are no competing interests.

### Author Contributions

- Yongbin Li conceived and designed the experiments, performed the experiments, analyzed the data, prepared figures and/or tables, authored or reviewed drafts of the paper, approved the final draft.
- Haowen Shi performed the experiments, analyzed the data.
- Haowei Zhang performed the experiments.
- Sanfeng Chen conceived and designed the experiments, contributed reagents/materials/analysis tools.

### DNA Deposition

The following information was supplied regarding the deposition of DNA sequences:

The nifB sequences described here are accessible via GenBank accession numbers MH202771.

The amyE sequences described here are accessible via GenBank accession numbers MH202772.

### Data Availability

The raw measurements are provided in Data S1.

### Supplemental Information

Supplemental information for this article can be found online at http://dx.doi.org/10.7717/peerj.6073#supplemental-information.

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
