# Peer review of "Amelioration of drought effects in wheat and cucumber by the combined application of super absorbent polymer and potential biofertilizer"

_PeerJ, doi:10.7717/peerj.6073_

## Round 0.1 · original submission · Minor Revisions

Please see the Reviewers' comments below and provide point by point responses how you have addressed their suggestions in your manuscript. I look forward to receiving your responses and the revised manuscript.

Reviewer 1 ·

Basic reporting

Regarding the manuscript, it is clear and well organized. The language is appropriate and the presentation of the results is nicely summarized in figures and tables.

Experimental design

The main concern for the reviewer is the number of days that was considered for harvesting. It is important to evaluate full-life cycles; however, short-term studies are equally valid with a strong justification. The reviewer considers that such justification is missing in this article.

Validity of the findings

The discussion is clear and cites relevant literature that ratify the results found in the investigation. The narrative is clear and easy to understand. The reviewer suggests incorporating a few references from more recent investigations in the discussion (2018

Additional comments

Specific comments

Language
- The article is well written and with a structure that is easy to follow. It makes the reading fluid and easy.
- Throughout the text, there are various cases of missing commas after a list when there is a “and” at the end. Example: Line 47: “solubilization and indole-3-acetic acid (IAA) production” should be written as “solubilization, and indole-3-acetic acid (IAA) production”.
- Lines 18 and 188, please replace manufacture’s with manufacturer’s
- Line 239, letter L from L-56 is missing.

Abstract
- The abstract summarizes accordingly the findings of the investigation.
- Line 23 reads “both biofertilizers amended with SAP could tremendously promote germination”. The reviewer suggests using a word that can be easily quantified or omit the word tremendously.

Introduction
- Line 38 reads “Traditional agricultural practices have fatal negative influences on…” Are fatal and negative synonyms?
- Lines 60-61 “However, some biofertilizers failed to promote plant growth in the field” Could you provide some examples/references for this statement?
- Line 75, please define Superab A200
- Lines 76-78 “Most studies only focused on the application effect of SAP alone or mixed with chemical fertilizer, while that of synergistic use of SAP with biofertilizer were rarely reported, particularly under water deficit conditions” Please provide examples of reports with some results. If there are none, write a statement such as “To the authors’ knowledge, there are no reports…”
- Line 83, please define all elements (or none).
- Line 91, could you justify why “greatly different crops” were selected? Why specifically wheat and cucumber?

Materials and Methods
The methods are overall clear and well written. Specific details to be modified include:
Bacterial strains and potted plant soil
- Table 1 describes L56 as obtained from storage in the laboratory. Was this purchased in the past? Who is the manufacturer? Where was it obtained from? Please provide more detailed information.
- Line 106, please define O.M.
Plant culture and collection
- In line 124 (and all treatments including SAP), if the weigh of SAP is ignored, how can this experiment be replicated? Could you provide more information on the treatment preparation?
Viability assessment of inoculants in rhizosphere soil
- Line 141, why 14, 19, 24, 29, 34, 39 and 44 days after sowing were selected?
Sample collection and preparation
- Why were the samples harvested at day 44? (line 159)
Determination of free proline content and total soluble sugars (TSS) in plant leaves
- What is the relevance of measuring free proline and TSS. Could you include a short sentence to justify these assays?

Results
The results are clear and nicely summarized in figures and tables. The reviewer suggests to increase the font size in axis names and legends in figures, as well as showing the standard error bars as ±.

Discussion
The discussion is clear and cites relevant literature that ratifies the results found in the investigation. The narrative is clear and easy to understand.
- Line 345 “Similar results have been reported (Siddikee et al., 2016).” In which plants?
- The reviewer suggests to incorporate a few references from more recent investigations in the discussion (2018).

Conclusions
The conclusions are clear and the take-home message is nicely summarized in the paragraph “Our results indicate that SAP addition in biofertilizer is a good strategy for improving the efficiency of biofertilizer, especially in the areas suffered from long-term drought stress.”
- In line 388, the reviewer suggests to avoid the use of the word tremendously

Annotated reviews are not available for download in order to protect the identity of reviewers who chose to remain anonymous.

Reviewer 2 ·

Basic reporting

Even though the writing is overall of good quality, the text needs to be improved. In some cases the use of words is not the appropriate, even though the spelling is correct (e.g. lines 312, 390). Be objective, provide measurable information, facts. Use articles and connecting words appropriately. The introduction and literature cited are relevant as well as the Figures provided. However, the quality of the images need to be improved as it is difficult to read the letters/words.

Experimental design

The research is within the scope of the journal and the question is well defined (can SAP help the proliferation of biofertilizers). The investigation includes high technical standards and the methods are described in detail.

Validity of the findings

Data is robust, statistically sound and controlled. However, I suggest to double check the error bars. They are relatively small for having only n=3. These may be alright, it is a comment as I am used to seeing bigger error bars in plant samples with such a small number of replicates. Speculation was kept to the minimum, the conclusion is linked to the original question and does not bring extra information at last. Also, SAP did improve the viability of the inoculants. However, is this due to an improvement in moisture content? Why was moisture not recorded? Did I miss it?

Additional comments

Even though the writing is overall of good quality, the text needs to be improved. Three main concerns: Double check the error bars (details above), Figures’ resolution, and explain why moisture was not recorded, if the whole point was to prove that SAP could help overcome drought issues for the survival of biofertilizers.
Various lines – “i.e.” is not needed in most cases.
Various lines – Check spelling (e.g. lines 27, 239, 453, etc.). Avoid starting sentences with the word “and”.
Various lines – Improve use of articles and other words such as “was, were, are”. E. g. line 77 …or mixed with “a” chemical fertilizer, while “studies on the” synergistic use of SAP with biofertilzer “are” rarely…
Various lines – Check for sentence fragments or statements that don’t belong. E.g. lines 83-84, 87, etc.
Various lines – Be objective, provide measurable data. E.g. in line 91 justify why wheat and cucumber were selected, if you state they are “greatly” different, point out those differences; line 117 “super clean bench” provide concise conditions; 137 specific details, 161 by how much, 371 what “great” roles, 388 how “tremendously”, etc.
Various lines – Be clear. E.g. in line 95 what do you mean by “generalize”? Improve clarity throughout the text. E.g. lies 265-275, 298-299 do you mean trend? 303-304, etc.
Figures and Tables: Double check statistics, image resolution, remove extra dot in Fig. 4 caption, Table 1 "storage in the lab" may not be appropriate, rearrange Table 5 column 6.

---

## Round 0.2 · accepted · Accept

Thank you for addressing the reviewers' comments and revising your manuscript. I look forward to publication of your article!

#